# Do xenophobic attitudes influence migrant workers' regional location choice?

**Tanja Buch**[1☯], **Carola Burkert**[2☯*], **Stefan Hell**[3☯], **Annekatrin Niebuhr**[1,4☯], **Anette Haas**[5]

**1** Institute for Employment Research, IAB Nord, Kiel, Germany, **2** Institute for Employment Research, IAB Hesse, Frankfurt, Germany, **3** Institute for Employment Research, IAB Rhineland-Palatinate-Saarland, Germany, **4** Empirical Labour Economics and Spatial Econometrics, Department of Economics, Christian-Albrechts-Universitaet zu Kiel, Kiel, Germany, **5** Institute for Employment Research, Nuremberg, Germany

☯ These authors contributed equally to this work.
\* carola.burkert@iab.de

## Abstract

This paper investigates whether xenophobic attitudes, as measured by the regional share of votes for right-wing parties and xenophobic violence, affect migrants' choices of where to live in Germany. We use a unique panel data set for the period 2004 to 2017 and apply fixed effects regression models and instrumental variable estimation to examine the relationship between anti-immigrant attitudes and immigration. Our results indicate that xenophobic attitudes tend to reduce regional labour immigration. However, evidence seems to be more robust for the support of right-wing parties than for xenophobic violence. Regarding heterogeneous effects across skill groups, the findings are ambiguous. While the immigration of skilled workers seems to be more sensitive to xenophobic violence, evidence is more robust for the share of right-wing votes in the case of low-skilled foreign workers. The strength of the adverse effect of anti-immigrant attitudes tends to increase with the local size of the coethnic community.

**Data Availability Statement:** All relevant data are within the paper and its Supporting Information files.

**Funding:** The author(s) received no specific funding for this work.

## 1. Introduction

Demographic change will have a significant impact on European labour markets soon. As baby boomers begin to retire, the supply of labour is noticeably decreasing [1]. An important approach to mitigate these demographic trends is immigration [2]. Thus, countries' and likewise regions' ability to attract workers from abroad is important for their economic prospects. However, we observe a significant variation in immigration both across countries [3] and between regions within countries [4, 5]. Often, the regions that experience a low level of immigration are structurally weak and affected most severely by demographic change [6].

At the same time, increasing international labour migration and high numbers of asylum seekers have sparked a heated debate about immigration in destination countries during which right-wing parties have gained importance and in turn have further fuelled the debate [7]. As with immigration, there is large variation in the support for right-wing attitudes both across countries and regions [8, 9].

**Competing interests:** The authors have declared that no competing interests exist.

If right-wing attitudes send a repelling signal to potential immigrants, they may impede economically desirable immigration. This might affect regions very differently because stereotypes and prejudices have a strong local dimension as they contribute to the community culture, which defines the broader societal traits and relations that shape places in terms of the prevailing mindset and overall way of life [10, 11]. Several studies show that negative attitudes towards immigrants affect their well-being and social integration in the host country [12–14]. These findings suggest that xenophobic attitudes might also influence migration decisions. Slotwinski and Stutzer [15] note that corresponding evidence is, however, scarce. A small number of previous studies focus either on the impact of anti-immigrant sentiments on migration flows between countries [16, 17] or on the internal migration of foreigners within the host country [18, 19]. Overall, the research points to a negative relationship between anti-immigrant attitudes and migrants' decisions to settle.

This paper investigates the effects of regional differences in xenophobic attitudes, measured by votes for right-wing parties in federal elections and xenophobic violence, on the immigration of foreign workers to German district regions. The electoral success of right-wing parties is used as a proxy for xenophobic attitudes as there is evidence that anti-immigration sentiments are highly correlated with right-wing party support [20] and that the results of national elections reflect local values and attitudes [21]. Our data on xenophobic violence should also indicate the extent of xenophobic attitudes in a region as it is based on regional information from the German security authorities, which records crimes with racist, xenophobic or anti-Semitic motives.

Our analysis focuses on labour migrants, i.e. we neither consider refugees nor student or educational migration. We expand the existing research in various ways. We provide first evidence on the relationship between xenophobia and immigration for Germany, which has become an important destination country for international migrants [22]. Furthermore, we contribute to a relatively new strand of literature that investigates the initial location choices of immigrants in destination countries [5]. Within this small group of studies, to our knowledge, only Bracco et al. [18] also consider the influence of anti-immigrant attitudes on immigrants' first destination choices in a country. We additionally examine whether this effect differs significantly across skill groups and between EU and non-EU nationals. Finally, we investigate whether the effect of xenophobic attitudes is influenced by the local size of coethnic communities.

Germany is particularly well suited to examining the influence of xenophobia on immigrants' choices of a region of residence. The number of immigrants as well as the electoral successes of right-wing parties and the number of right-wing violent acts increased significantly in the past two decades, albeit with large regional differences. This temporal and regional variation is an important prerequisite for investigating the effects of xenophobic attitudes on immigration.

The analysis rests on a unique panel data set for the period 2004 to 2017 that includes detailed regional information on labour immigration from abroad, election results, xenophobic violence, and various characteristics of German district regions, our spatial units of observations (see section 4.1 for details). We apply fixed effects regression models and instrumental variable (IV) estimation to account for unobserved regional characteristics that might affect immigration and the potential endogeneity of the key explanatory variables.

The paper is organized as follows. The next section provides descriptive evidence on the development of immigration and xenophobic attitudes in Germany in the past two decades. In section 3, we survey the literature on the impact of hostile attitudes on the settlement decisions of migrants and derive the hypotheses to be tested in our empirical analysis. Section 4 describes the data, variables, and empirical strategy. We discuss the results in section 5. Section 6 concludes.

## 2. Development of immigration and xenophobia in Germany

In the past two decades, the number of immigrants to Germany has risen considerably but not linearly (see Fig 1). In 2001, approximately 685,000 foreigners immigrated to Germany. Until 2009, the annual immigration was lower. After 2009, annual figures significantly increased. This was mainly due to two changes: the complete freedom of movement for workers from Eastern European countries as result of the eastward enlargement of the European Union (EU) and the liberalization of labour migration from non-EU countries [23]. After peaking in 2015 with the sharp increase in refugee immigration, the number of immigrants declined but remained approximately 50% higher than it was at the beginning of the century.

Large regional disparities characterize immigration patterns, mainly between East Germany and West Germany. While around 85% of Germany's residents live in West Germany (including Berlin), over the last two decades 90% to 94% of immigration has been concentrated in West Germany [5]. In our data, the immigration rate in 2017, defined as migrant employees per 1,000 employees in a region, varies from a minimum of 1.4 (Bautzen/Saxony/East Germany) to a maximum of 24.1 (Offenbach/Hesse/West Germany).

Similar to the development of immigration, support for right-wing parties has increased steadily in recent years (see Fig 1). Their share of votes in federal elections was relatively low until 2009, but rose to 6.5% by 2013 and doubled to 13.2% in the 2017 election. Among the right-wing radical parties (selection explained in section 4.1), the Alternative for Germany (Alternative für Deutschland, AfD) dominates with a share of 96% of all right-wing votes in the 2017 federal election. Although xenophobia is undoubtedly a country-wide problem, there are also large regional differences. For example, in the 2017 federal election, the minimum vote share was 5.2% in Muenster, a university city in West Germany, and the maximum was 37.6% in the East German rural region Saechsische Schweiz-Osterzgebirge. In Germany's most recent federal election in 2021, the percentage of votes for the AfD was 19.1% in East Germany, more than twice as high as the 8.2% in West Germany. Reaching a share of nearly

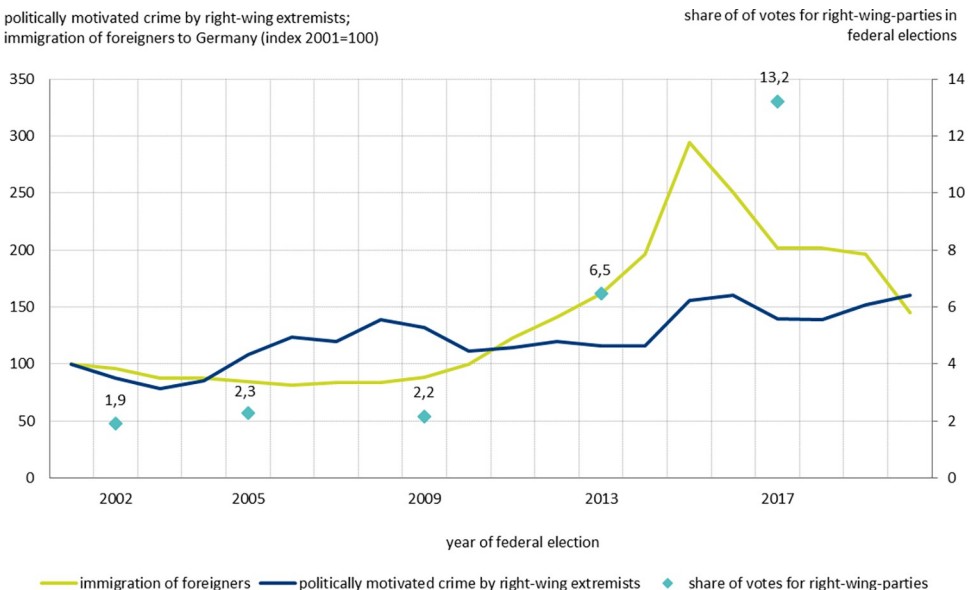

**Fig 1.** Immigration of foreigners to Germany 2001–2020, politically motivated crime by right-wing extremists (both left axis, index 2001 = 100), and share of votes for right-wing parties in federal elections (right axis). Source: Federal Statistical Office, State Offices of Criminal Investigation, the federal returning officer, own calculations.

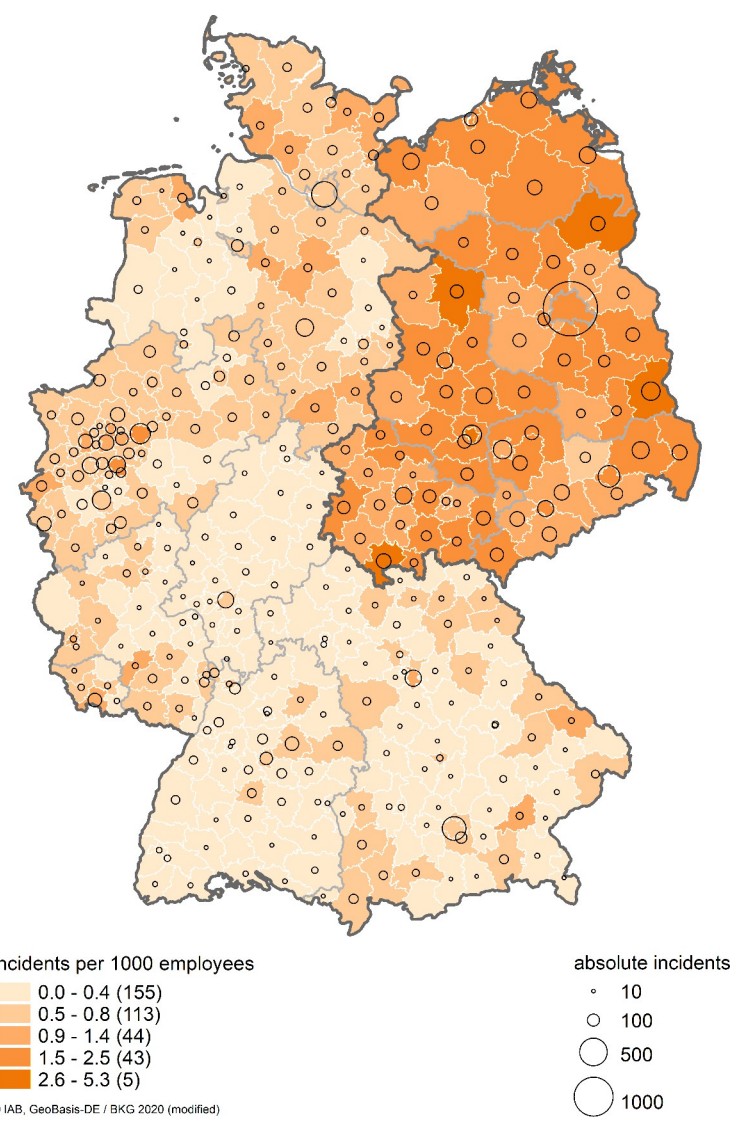

incidents per 1000 employees

absolute incidents

- 0.0 - 0.4 (155)
- 0.5 - 0.8 (113)
- 0.9 - 1.4 (44)
- 1.5 - 2.5 (43)
- 2.6 - 5.3 (5)

© IAB, GeoBasis-DE / BKG 2020 (modified)

- 10
- 100
- 500
- 1000

**Fig 2. Politically motivated crime by right-wing extremists in German district regions—absolute number of incidents and incidents per 1,000 employees in 2017.** Notes: On the map, the absolute number of incidents is marked by black circles, more incidents are indicated by a larger radius. The relative frequency of incidents is indicated by colouring: the darker the area, the higher is the number of incidents relative to the number of employees. Source: Federal State Criminal Police Offices, own calculations.

25% of votes, the AfD was even the strongest party in the eastern German federal states of Saxony and Thuringia.

Finally, politically motivated crime by right wing extremists has risen in recent years (see Fig 1). After a slight decline at the beginning of the first decade of this century, hate crime rose significantly, particularly after the sharp increase in refugee immigration in 2015. Fig 2 shows right-wing politically motivated crime in German district regions in absolute and relative terms in 2017. The highest numbers of incidents are observed predominantly in large cities, such as Berlin, Hamburg and Munich. In relative terms, measured by the incidents per 1,000 employees, we again see substantial differences between East Germany and West Germany, with much more assaults per capita in the East (see [24, 25] for similar findings). The

minimum value (0.07) is found in the district region Freudenstadt in Baden-Wuerttemberg in West Germany and the maximum value (5.3) is found in Hildburghausen in Thuringia in East Germany. For a discussion of the reasons why right-wing extremist attitudes find strong support especially in East Germany, see, e.g., [26, 27].

## 3. Theoretical considerations and empirical evidence

Two strands of literature address the interaction between immigration and locals' attitudes towards migration. The first strand focuses on the question of whether immigration affects attitudes towards immigration with two competing theoretical approaches: according to the 'racial threat hypothesis' [28], the population majority, when confronted with a growing minority fraction, is likely to impose social controls as they try to defend their dominant political, economic and social positions. In contrast, the 'contact theory' [29, 30], suggests that migration increases contact between members of majority and minority groups and thereby reduces racial tensions by providing reliable information on minority behaviours, values, lifestyles and experiences. Empirical research does not provide a clear-cut picture of this issue. While some studies find evidence for the racial threat hypothesis (e.g., [9, 31]), others support contact theory (e.g., [32]). Gorinas and Pytliková [16] provide a comprehensive overview of the factors that positively or negatively correlate with natives' attitudes towards immigrants and thoroughly discuss the underlying mechanisms.

For our study another strand of literature is of primary interest which examines the opposite relationship, i.e., whether locals' attitudes towards migrants influence immigration. This specific issue is part of an extensive literature that deals more generally with factors that impact the location choices of migrants [33]. Theoretically, migrants choose the destination where the expected utility, considering migration costs, is highest [34, 35]. Different location characteristics influence the expected utility and might act as push or pull factors repelling or attracting potential migrants. Many studies emphasize the role of economic and labour market conditions such as regional wage differentials [36] and unemployment disparities [37] or differences in employment growth [38]. Moreover, the housing market [39] and amenities such as a pleasant climate [40] or recreation areas [38] are supposed to impact regional quality of life and thereby migration decisions. The same applies to expected social well-being in a region [41]. Most studies that analyse non-employment-related factors as drivers of migration decisions focus on internal migration while the importance of non-income motives for international migration decisions has been less studied. However, Manchin [42] and Heider et al. [43] confirm the relevance of local amenities for international migration intentions.

As amenities may attract migrants, disamenities such as (street) crime [44] or environmental burdens [45] may repel them. We assume that xenophobic attitudes might act as a disamenity, as fears of rejection or even violence from natives might reduce the expected social well-being of migrants. Knabe et al. [12] show with German data that the life satisfaction of immigrants is significantly reduced as regional right-wing extremism increases. This effect is strongest for highly skilled immigrants. Steinhardt [14] explores the effect of xenophobic violence events in the early 1990s in West Germany and finds an overall decline in subjective well-being for the Turkish community. Analysing the feeling of welcome and belonging of migrants in Berlin and Amsterdam, Cichocka [46] finds a positive effect on the social well-being of migrants. Consequently, regions with more hostile attitudes towards foreigners should receive, ceteris paribus, smaller migrant inflows. Departing from these arguments, our first hypothesis is given by:

H1: The immigration rate of a district region declines as xenophobic attitudes in the region increase.

Skilled people seem to be more sensitive to xenophobic attitudes [12]. This could be due to a stronger intention of better educated immigrants to identify and integrate with their host country, resulting in stronger disappointment if the native-born population reveals negative perceptions of immigration [12]. Moreover, skilled people tend to have more regional job opportunities to choose from [47]. We therefore presume that the negative relation between the level of hostile attitudes in a region and its immigration rate should especially be true for skilled immigrants.

H2: The negative effect of xenophobic attitudes on immigration rate is stronger for the skilled compared to the unskilled immigrants.

There are differences in the freedom to choose where to live between migrant workers from the EU and those from third countries. Labour migrants from third countries have no free access to the German labour market and in general their residence permit is linked to a labour contract. In contrast, labour migrants from EU-countries face no restrictions in their location choice (full freedom of movement for workers). Third country nationals might therefore be less able to avoid a xenophobic environment than EU-nationals when choosing a region of residence. Additionally, media coverage of social and political phenomena tends to decline with increasing regional distance [48, 49]. Therefore, we might expect that EU foreigners react more sensitively to revealed anti-immigrant attitudes, because they are better informed than people from more distant third countries.

H3: The negative effect of xenophobic attitudes on immigration rate is stronger for EU nationals compared to non-EU nationals.

The migration decision may additionally be impacted by the size of the coethnic community in a region [50]. Migrants usually have contact with coethnics already living in a potential destination region who share their experiences (see Radu [51] for an overview of the impact of social interaction on migration choices). Additionally, social media have become popular many-to-many channels of communication that make prospective migrants more informed about possible settlement destinations [52, 53]. In this way, migrants can obtain information about local anti-immigrant attitudes via their ethnic community, which may affect their decision-making. Examining online mutual-help communities of Indian aspirant student migrants on Facebook and WhatsApp, Jayadeva [54] shows that fear of racism is an issue discussed when deciding to study in Germany. However, whether a large local community of coethnics amplifies or dampens the effects of xenophobic attitudes is an open question. On the one hand, it is conceivable that migrants are more likely to move to regions with right-wing natives if they assume that a large coethnic group offers mental support [55]. Showing that the experience of racism is lower in places of higher ethnic density, Bécares et al. [56] speak of a "buffering effect" of ethnic density.

H4a: Negative effects of xenophobic attitudes on immigration rates decline in absolute size as the local size of the own ethnic group increases.

On the other hand, a large coethnic group can collect and spread more information, e.g. via mass media [57]. If the information is about the hostile attitudes of the local population and warns against immigration, this may amplify adverse effects of xenophobic attitudes.

H4b: Negative effects of xenophobic attitudes on immigration rates increase in absolute size as the local size of the own ethnic group increases.

There is little empirical evidence on the impact of local anti-immigrant attitudes on migration behaviour. Some studies deal with this topic at the country level. Gorinas and Pytliková [16] take a cross-country approach and find a negative effect of anti-immigrant attitudes on

migration flows between OECD countries. Avdiu [47] confirms these results and emphasizes an effect especially on highly qualified people. Friebel et al. [17] look at the effect of xenophobic attacks in South Africa on the migration intentions of Mozambicans and likewise find negative effects. For Europe, De Coulon et al. [58] find adverse effects of hostile attitudes of natives on the intentions of migrants to stay in Italy.

More relevant for our study is research on the location decision of people from abroad within the destination country. Some studies provide evidence on the impact of majorities' hostile attitudes on internal migration. Henry [59] shows that the probability of African Americans settling in a specific city in the U.S. is significantly reduced by the level of race-based crimes and racially intolerant attitudes against them. For Europe, Damm [60] finds that the probability of refugees' relocating in Denmark increases with the percentage of right-wing votes in the latest local election. Bracco et al. [18] find that the election of a far-right Lega Nord mayor discourages immigrants from moving into the municipality. Schmutz and Verdugo [61] show with French data that the share of immigrants in the population grows faster in municipalities where a left-wing rather than a right-wing mayor has been elected. Slotwinski and Stutzer [15] investigate the internal migration of foreigners in Switzerland and show that the probability of moving to a municipality decreases with revealed reservations towards them. Additionally, for Switzerland, Bennour et al. [62] find that (mainly highly educated) immigrants are likely to favour municipalities where natives' attitudes and naturalization requirements are inclusive.

Hardly any information exists on the impact of xenophobic attitudes on the settlement structure of immigrants in Germany. Some qualitative research supports the idea of a negative impact of hostile attitudes on immigration. Notably, East German employers mention xenophobia and right-wing extremist violence as locational disadvantages. In a survey by Bussmann and Werle [63], companies in East Germany suspect that applicants' fears of right-wing extremist attacks explain 20% of applicants' job rejections.

In summary, theoretical considerations as well as empirical findings point to a negative impact of local xenophobic attitudes on migrants' decisions to take a job and settle down. However, the current evidence rests on very few studies, and none makes use of German data. Thus, it is an open question for research whether the supposed fear of applicants regarding right-wing extremist attacks mainly in East Germany in fact explains the settlement structure of labour migrants and might thereby tighten regional skilled labour shortages.

## 4. Data and empirical strategy

### 4.1 Data

The analysis of regional immigration rests on a panel data set that covers the period from 2004 to 2017. We use the Integrated Employment Biographies (IEB) of the Institute of Employment Research (IAB) to generate our regional immigration data. The IEB provide detailed information on all workers covered by the social security system in Germany. For more information on the IEB data set, see [64].

As the IEB do not provide information regarding migrants' country of birth or year of arrival, we follow Tanis [5] and identify migrant workers using information on nationality and their first appearance in the IEB. The location choice is where the first residence in Germany was registered in the IEB and the first status that we observe in the IEB must include a period of at least three months in regular employment. In doing so, we leave out other forms of migration and ensure that we focus on labour migration. We thereby inter alia exclude refugees from the data because they do not have immediate access to the labour market and their

first status in the IEB is a social benefit receipt. Our definition of migrant workers also excludes young individuals who enter the country to start an apprenticeship.

To ensure that foreign employees who grew up in Germany are excluded from the analysis, we restrict the sample to first IEB employment spell of workers in the age group between of 28–62 years, as we assume that the IEB periods of younger foreign workers likely indicate that they grew up in Germany.

## 4.2 Dependent variable

The individual immigration events are aggregated at the district region level. We use district regions because they are comparable spatial units of analysis. For this purpose, the 400 districts and independent cities in Germany, small administrative regions which generally have a population of 150,000 to 800,000 inhabitants, are aggregated to 360 district regions by combining independent cities with fewer than 100,000 inhabitants with their surrounding districts. The IEB data includes approximately 1,729,970 individual immigration events, i.e. first employment notifications of foreign workers in the IEB between 2004 and 2017 that meet all conditions set out above. We use a dependent variable that is similar to a measure proposed by Mitze [65] and define the immigration rate $imr_{it}$ of region $i$ in year $t$ relative to the number of workers residing in region $i$ in the previous year ($workpop_{it-1}$):

$$\ln imr_{it} = \ln\left[\frac{immig_{it} + workpop_{it-1}}{workpop_{it-1}}\right] \quad (1)$$

The variable indicates the relative change in the regional workforce caused by immigration. A value of 0.01 thus means that immigration gives rise to a 1% increase in the regional workforce (10 immigrants per 1,000 workers). We also generate migration data for different groups of foreign workers and differentiate by skill level, nationality, and rules regarding access to the German labour market (EU, non-EU). Related to skill groups, we differentiate between skilled immigrants, who have either a university degree or a formally certified completed apprenticeship training and the remaining workers, comprising unskilled migrants and those with unknown qualifications. In an extended model, we consider effects of coethnic communities on the immigration rate of specific nationalities, focusing on the ten largest immigrant groups in Germany (see section 4.2, Eq (3)).

## 4.3 Main explanatory variables: Indicators for xenophobic attitudes

We use two indicators to measure xenophobic attitudes: first, xenophobic violence and second, electoral votes for right-wing parties. Information on right-wing violent political action comes from the statistics on politically motivated crime that are registered and collected by the Federal State Criminal Police Offices (Landeskriminalaemter). The responsible authorities categorize the motive behind an incident in case the local police authorities suspect a political motive. They record racist, xenophobic, anti-Semitic and other crimes with a right-wing motivation. The statistic includes violent right-wing extremist crime such as murder, assault and fire attacks as well as non-violent incidents, e.g., right-wing extremist propaganda, threat and coercion (see [66]). Annual figures are available for all district regions, albeit for different periods.

It is being discussed whether the recording of politically motivated crime from the right is underrepresented [67] and therefore subject to measurement problems. This is possible because for the statistics on politically motivated crime, criminal offences are judged to be politically motivated or not immediately at the start of police investigations. If the assessment turns out to be incorrect during the course of the investigation, the crime data is not

subsequently revised. However, Entorf and Lange [68] argue that the official hate crime reports are less prone to under-reporting than surveys or newspaper data and also provide corresponding empirical evidence. But altogether, the correlation between official police statistics and other data seems to be high. Braun and Koopmans [69], for instance, report a correlation of 0.82 between yearly aggregates of newspaper data and right-wing incidents documented by the police. There are several studies that make use of the statistics on right-wing politically motivated crime to investigate right-wing hate crime [e.g., 66, 68].

Information on election results of right-wing parties in federal elections in the period from 2002 to 2017 is based on the second votes, which decide the allocation of seats in Parliament. There is evidence that anti-immigration attitudes are a strong predictor of right-wing party support [70–72]. We define parties as "right-wing" on a political spectrum from right-wing populist to far right according to the Federal Agency for Political Education in Germany (Bundeszentrale für politische Bildung). The parties NPD, REP, DVU, DIE RECHTE, pro Deutschland, AfD, Offensive D, Ab jetzt. . . Demokratie durch Volksabstimmung, Pro DM/DM, BüSo, and BfB are classified as right-wing. We exclude those right-wing parties that reached less than 10,000 second votes in sum from all federal elections from 2002 to 2017. For our analysis, we aggregate the regional voting shares of all considered parties at the district level for every federal election. Most of the parties did not compete in every election or in all regions.

Previous studies do not offer a clear picture of the correlation between the occurrence of xenophobic attacks in a region and support for right-wing parties. In some studies, the variables are negatively correlated, which is attributed to an opportunity effect. Accordingly, people are less likely to commit right-wing extremist attacks where right-wing parties represent a relevant force within the political system and thus an alternative way of exerting pressure on political elites [73]. Other studies do not find any significant relationship [69]. For Germany, Jäckle and König [74] show that the probability of attacks on refugees positively correlates with the local strength of right-wing parties. This finding was, however, not confirmed in a replication study [75]. Both factors, xenophobic attacks and election results, might reflect different aspects of xenophobic attitudes and thus influence migration decisions differently.

### 4.4 Coethnic communities

To account for the effect of ethnic communities on the choice of a region of residence, we use the population share of an individual's own nationality as a proxy. We focus on the largest groups of foreign workers who reside in Germany (Bulgarian, Chinese, Greek, Indian, Italian, Croatian, Polish, Romanian, Russian, Spanish, Syrian, Turkish, Hungarian). They account for almost 59% of all immigration events in our IEB data set. The average share of the largest groups varies between 12.3% for the Polish immigrants and 1.7% for immigrants from Spain. More detailed information on the size of the different groups is provided in Table A8 in S8 Table.

To account for the potential influence of local coethnic communities on the effect of anti-immigration attitudes we extend our basic empirical model by including interaction terms between the local share of one's own group and the corresponding xenophobia measure.

### 4.5 Control variables

To control for the impact of various push and pull factors (see section 3), we use variables reflecting location characteristics in terms of economic conditions, labour market, amenities and disamenities (see Table A1 in S1 Table for a detailed description of the variables and its data sources and Table A2 in S2 Table for summary statistics).

## 4.6 Sample size

The data set is an unbalanced panel although we have annual information on immigration and most explanatory variables. The unbalanced structure of the data is primarily due to the indicators for xenophobic attitudes. First, federal elections do not take place every year. As immigration data is available for the period 2004 to 2017 and all regressors enter with a time lag, we consider the election results in 2005, 2009 and 2013 in our analysis. Second, information on politically motivated right-wing crime is not available annually for the entire period in all district regions because federal states started to report the figures at different points in time. The number of observations in the regression analysis therefore differs depending on whether we investigate the effects of xenophobic violence (2,772 observation in the main model, fixed effects IV specification) or electoral votes for right-wing parties (1,050 observations, see also notes below Table A1 in S1 Table for details). Sample size increases for the extended model, because we regress an annual immigration rate for the ten largest nationality groups in every district region on the explanatory variables (30,354 and 10,471 observations, respectively).

## 4.7 Empirical strategy

To investigate how xenophobic attitudes affect the immigration rate, $imr_{it}$, we apply a regression model given by:

$$\ln imr_{it} = \alpha + \beta \ln\left(\frac{xenophobia_{it-1}}{xenophobia_{Gt-1}}\right) + \sum_{l=1}^{L} \gamma_k \ln\left(\frac{z_{it-1}^l}{z_{Gt-1}^l}\right) + \delta_i + \theta_t + \varepsilon_{it} \qquad (2)$$

The pivotal explanatory variable is $\left(\frac{xenophobia_{it-1}}{xenophobia_{Gt-1}}\right)$, which is based on either the number of xenophobic crimes per worker living in region $i$ or the share of votes for right-wing parties. The indicators for xenophobic attitudes are included as logarithms of ratios that measure the relative deviations of the local conditions from the respective German average $xenophobia_{Gt-1}$ excluding the region under consideration (see [65]). Moreover, we consider numerous control variables $\left(\frac{z_{it-1}^l}{z_{Gt-1}^l}\right)$, which include regional labour market conditions and different local (dis) amenities, again relative to the respective German average excluding the region under consideration. Using logarithms of the dependent and the explanatory variables in the model is a rather common approach in migration research (see, e.g., [16, 43, 65]). This specification has also the advantage that all coefficients can be interpreted as elasticities, i.e. they give the percentage change of the immigration rate for a change of the corresponding explanatory variable by 1%. All explanatory variables are included as time lags ($t$−1).

The specification includes region-specific effects $\delta_i$ that capture the impact of unobserved time-constant determinants of the regional immigration rate, $\theta_t$ denotes time effects, and the error term is given by $\varepsilon_{it}$. We estimate the model given by Eq (2) for all immigrants, but also examine whether the effect differs across skill groups and between EU and non-EU nationals.

In the extended model, we differentiate between ethnic groups to investigate the influence of local coethnic communities on the choice of a region of residence. This means we calculate an immigration rate $\ln imr_{it}^g$ for different nationality groups $g$, add the share of one's own ethnic group ($group_{it-1}$) and also include an interaction term between the xenophobia measure and the coethnic community indicator in the model. The latter is supposed to capture a

potential influence of local ethnic communities on the impact of xenophobia, denoted by $\tau$:

$$\ln imr_{it}^g = \alpha + \beta \ln\left(\frac{xenophobia_{it-1}}{xenophobia_{Gt-1}}\right) + \sum_{l=1}^{L} \gamma_k \ln\left(\frac{z_{it-1}^l}{z_{Gt-1}^l}\right) + \delta_i + \theta_t + \rho \ln group_{it-1}$$
$$+ \tau \ln\left(\frac{xenophobia_{it-1}}{xenophobia_{Gt-1}}\right) \cdot \ln group_{it-1} + \varepsilon_{it} \qquad (3)$$

A major challenge of our analysis concerns the potential endogeneity of xenophobic attitudes leading to a biased estimate of the parameter $\beta$ as discussed by Gorinas and Pytliková [16]. There are two main causes of endogeneity in this context. First, there might be omitted influential factors that impact the immigration rate and correlate with xenophobic attitudes. We apply a fixed effects model to account for the influence of unobserved regional characteristics and year-specific shocks. An extensive set of time-variant controls and fixed effects should help to minimize the risk of an omitted variable bias.

The estimates of the effect of xenophobic attitudes might nevertheless be biased if there are relevant time-varying determinants of the immigration rate that we do not observe or may arise from reverse causality. As discussed in section 3, anti-immigrant attitudes may influence immigration, but at the same time, immigration might strengthen (or weaken) xenophobia. To generally reduce the risk of reverse causality, we treat all influential factors as predetermined and lag time-variant explanatory variables in the model by one year, an approach also applied by Gorinas and Pytliková [16]. Furthermore, we apply fixed effects instrumental variable estimation to deal with the potential endogeneity of xenophobic attitudes.

Finding valid instruments for our migration model is particularly demanding because the first stage is also a fixed effects model. This means that the first stage uses within variation only to generate exogenous variation in xenophobic attitudes and leaves unused the cross-sectional variation in the data, which frequently makes up the major part of information available in regional datasets. Based on a review of the corresponding literature, we identify four regional characteristics as possible instrumental variables that should provide exogenous variation in xenophobia. The first instrument, the local supply of vocational training (instrumental variable i) acts as a proxy for employment opportunities for young people and more generally for limited opportunities and economic development prospects that people face primarily in rural areas and medium-sized and small cities as argued by Dijkstra et al. [76]. They coin the term 'geography of discontent' to describe the unhappiness experienced by people living in regions being characterized by economic stagnation and low productivity which may lead them to use elections to 'rebel against the feeling of being left behind' [77] and to vote for populist right-wing parties. Andersen and Zimdars [78] show that extreme right voters in Germany tend to perceive their own economic situations as bad. However, it is important that the availability of training positions does not directly affect the immigration of workers from abroad because the instrument should only influence immigration via its effect on xenophobic attitudes. We assume that this condition of the IV approach is fulfilled as the purpose of the training positions is training of graduates and not gainful employment. For the majority of the apprentice positions, one cannot make a living from the low apprentice wages. At the same time the supply of vocational training should be a good indicator for economic development prospects of the region and a factor that might significantly affect anti-immigrant attitudes.

The second instrument (ii), the share of low-skilled among the foreign workers, is based on a labour market competition argument put forth, for instance, by Edo et al. [31]. They argue that labour market concerns constitute one major channel through which immigration might affect natives. Economic models suggest that the labour market outcomes of natives with the same skills as foreign workers are adversely affected by immigration [79]. However, recent

studies indicate that this might only apply to low-skilled natives and low-skilled immigrants, while native workers tend to benefit from high-skilled immigrants through skill complementarities and innovation-enhancing effects [80]. Accordingly, natives with lower skills compete with immigrants for similar jobs and are therefore particularly sceptical about the immigration of low-skilled migrants. Findings by Edo et al. [31] confirm that the support for far-right parties increases primarily with low-skilled immigration. There is also some evidence suggesting that far-right voters are frequently low-educated [78, 71] or suffer from poor labour market prospects [78, 81]. Thus, a high share of low-skilled among the foreign workers living in the local labour market is supposed to increase anti-immigrant attitudes and should at the same time not directly affect the immigration rate of the district region. This is also supported by the results of reduced form regressions in Tables A5a and A5b in S5 Table which we discuss in detail in the S5 Table.

Regarding instrumental variable (iii), the population share of the foreign population lagged by 9 years, Edo et al. [31]. provide empirical evidence that immigration in general increases support for far-right politicians in France (see for similar results with Italian data [9]). However, from a theoretical perspective it is not clear whether the share of the foreign population affects xenophobic attitudes positively or negatively (see section 3). Hangartner et al. [82] provide an in-depth discussion of the different theoretical arguments.

With respect to the validity of the population share of the foreign population lagged by 9 years (covering the period 1995 to 2008) it is important to note that the share of the foreign population lagged by only one year (period 2003 to 2016) is included as an explanatory variable in the main model. The latter should capture the direct effects of networks or migration-specific amenities on the immigration rate, while the former should influence immigration only through its impact on xenophobic attitudes. We use a long lag of the share of foreign population as instrument because this approach assumes that it takes time for the corresponding demographic changes to influence attitudes and behaviour since people need time to realize that these changes are permanent. It is also likely that it may require some time to become acquainted with immigrants or to radicalize. Based on results of reduced form models, we assume that exclusion of the share of the foreign population lagged by 9 years from the immigration model is valid because it does not capture the effects of the contemporaneous share of the foreign population. A more detailed discussion of the reduced form models and results are provided in the S5 Table.

For instrumental variable (iv), the longitude of the region centre, Krueger and Pischke [24] show that the incidence of xenophobic crime is higher in East German regions and increases with distance from the former Iron Curtain. They argue that districts along the Polish border might be more insulated and therefore less exposed to immigrants and Western influence. We use the longitude of the district centre to capture these differences which likely reflect regional disparities in societal values and institutions. We interact the longitude with the instruments (ii) and (iii) and thus allow the effect of the exposure to (low-skilled) immigrants to vary with the location of the region. An increase in the share of (low-skilled) foreigners might therefore give rise to a stronger increase in xenophobic attitudes in the far eastern district regions.

To be valid, the instruments must be relevant and exogenous. They should correlate with xenophobic attitudes, must be uncorrelated with the error term $\varepsilon_{it}$ in the migration models (2) and (3), and should influence the immigration rate only through their effect on xenophobic attitudes. Apart from our discussion of theoretical arguments above, we provide evidence based on different standard econometric tests and on reduced form regressions to check the intuition behind our instruments and their validity. All results are discussed in detail in the S5 Table.

## 5. Regression results

Table 1 summarizes the regression results for the effect of right-wing votes on the immigration rate of different migrant groups. We focus on the fixed effects model (upper panel) and the IV estimates (lower panel). All models include time-varying explanatory variables and region fixed effects to control for observed and unobserved factors that might influence the choice of a residential area by immigrant workers. The results for all control variables are displayed in Table A3 in S3 Table. We refrain from a discussion of these results here, but provide a summary of the main findings together with Table A3 in S3 Table. To test hypotheses H1, we examine the average effect of right-wing votes on all immigrants in column (1) but also consider heterogeneous effects across skill groups (columns 2 and 3) and EU citizens versus non-EU citizens (columns 4 and 5) to examine the hypotheses H2 and H3 (see also Table 4).

The unconditional correlation between the share of right-wing votes and the immigration rate is negative (-0.00104) and highly significant. In this simple regression model, we include only the relative share of right-wing votes and time effects (see Table A6 in S6 Table). Including covariates and region-fixed effects slightly reduces the coefficient that gives us the average effect for all immigrants. The estimate in column (1) in Table 1 indicates that an increase in the share of right-wing votes by 1% relative to the remaining country reduces, on average, the immigration rate by 0.00094 percentage points or, in other words, a decline in immigration of 0.009 immigrants per 1,000 workers who reside in the region. Applying the range of votes in

**Table 1. Effects of right-wing votes on regional immigration rates.**

| | | | Fixed effects models | | |
|---|---|---|---|---|---|
| | (1) | (2) | (3) | (4) | (5) |
| | All immigrants | Skilled | Unskilled/unknown qualification | EU | Non-EU |
| Share of right-wing votes | -0.00094*** (0.00031) | -0.00025 (0.00022) | -0.00128* (0.00073) | -0.00044 (0.00030) | -0.00007 (0.00006) |
| N | 1,050 | 1,050 | 1,050 | 1,050 | 1,050 |
| $R^2$ overall | 0.11 | 0.05 | 0.05 | 0.06 | 0.37 |
| $R^2$ within | 0.82 | 0.81 | 0.71 | 0.78 | 0.66 |
| $R^2$ between | 0.22 | 0.11 | 0.09 | 0.13 | 0.43 |
| | | | Fixed effects instrumental variable estimation | | |
| | All immigrants | Skilled | Unskilled/unknown qualification | EU | Non-EU |
| Share of right-wing votes | -0.00764*** (0.00204) | -0.00569*** (0.00147) | -0.01283*** (0.00386) | -0.00569*** (0.00170) | -0.00084** (0.00032) |
| First stage | | | | | |
| IV supply vocational training | | | -0.00642*** (0.00193) | | |
| IV low-skilled foreign workers | | | 0.05683*** (0.01408) | | |
| Kleibergen–Paap LM test (p value) | | | 22.75 (0.0000) | | |
| F Test of excluded instruments | | | 13.14 | | |
| Hansen J test (p value) | 0.6129 | 0.2696 | 0.6910 | 0.5794 | 0.3362 |
| Anderson–Rubin Wald test (p value) | 30.51 (0.0000) | 34.08 (0.0000) | 20.64 (0.0000) | 20.29 (0.0000) | 11.57 (0.0031) |
| N | 1,050 | 1,050 | 1,050 | 1,050 | 1,050 |

All models include time-varying explanatory variables and region- as well as time-fixed effects to control for observed and unobserved factors. Robust standard errors in parentheses are clustered at the region level

* $p < 0.10$

** $p < 0.05$

*** $p < 0.01$.

Notes: IV supply vocational training: lagged (1 year) ratio of vocational training positions to graduates demanding training. IV low-skilled foreign workers: lagged (2 years) share of low-skilled among foreign workers interacted with longitude.

regions from the federal election in 2017 (32.4 percentage points, see section 2), this translates into a difference in the immigration rate between two corresponding district regions that differ only in the share of right-wing votes of 0.18 [(ln(37.6/5.2) x 0.0009) x 100] percentage points or approximately two foreign workers per 1,000 employees in the local labour market.

The average effect seems to be driven by the group comprising the unskilled workers and those with an unknown skill level. The effect of our pivotal variable is larger in absolute terms for this group, pointing to a decline of 0.00128 percentage points in the immigration rate if the election result of right-wing parties increases by 1% relative to all other district regions. Again, applying the range of votes at the district level from the election in 2017, the difference in immigration rates amounts to 2.5 immigrants per 1,000 residents. The effects for other sub-groups are clearly smaller and not precisely estimated.

The lower panel of Table 1 summarizes the results of the IV estimations (Table A3 in S3 Table shows the complete first-stage regression). Different tests on the validity of the instruments (local supply of vocational training, share of low-skilled foreign workers interacted with longitude) indicate that we cannot reject that they are exogenous and relevant. We discuss these results on instrument validity in more detail in the S3 Table.

The IV estimates reinforce the evidence of a dampening influence of xenophobic attitudes on immigration in the upper panel of Table 1. The effect of right-wing votes becomes larger in absolute terms, pointing to an upwards bias of the fixed effects estimates, which might be due to reverse causality, i.e., immigration affecting election results. The average effect across all immigrant groups amounts to 0.00764 percentage points, implying that immigration differs, ceteris paribus, by 15.1 foreign workers per 1,000 residents between the district regions showing the strongest and weakest support for right-wing parties. Our results for right-wing votes are thus in line with the literature that points to an adverse impact of xenophobia on immigrants' well-being and migration behaviours [15, 58, 60].

Evaluating the relatively low size of the effect, we need to consider that altogether, the level of immigration is low. The average annual immigration rate in our data set amounts to merely 4.3 immigrants per 1,000 inhabitants (see Table A2 in S2 Table). An increase in right-wing votes by one standard deviation relative to the remaining country reduces the immigration rate by 0.46, i.e., approximately 11% of the average rate if we use the IV estimate in Table 1 (-0.00764).

Apparently, immigration from other EU countries is more severely affected than the migration behaviour of non-EU citizens. However, the skill level of the immigrants seems to be more important regarding the heterogeneous effects of xenophobic attitudes. While we identify statistically significant effects for both skill groups, the effect for unskilled workers is also economically meaningful (25.4 workers per 1,000 inhabitants for the range of election results in 2017). Thus, regarding the impact of election results, our findings do not confirm earlier studies [12, 47]. that point to more significant effects of xenophobia on highly educated migrants.

Table 2 shows the regression results for the rate of xenophobic violence, differentiating again between fixed effects models (upper panel) and fixed effects IV results (lower panel). We refrain from a detailed discussion of the different tests applied to check the relevance and exo-geneity of the instrumental variables. The findings suggest that the instruments are valid (see information in S5 Table). Table A4 in S4 Table shows results for all controls and the complete first-stage regression.

There is a negative correlation between immigration and xenophobic violence. However, we detect a great decline in the correlation when comparing the unconditional correlation between right-wing violence and the immigration rate (-0.00151) and the fixed effects estimates. Neither the average effect in column (1) nor the estimates for different subgroups are

**Table 2. Effects of xenophobic violence on regional immigration rates.**

| | Fixed effects models | | | | |
| --- | --- | --- | --- | --- | --- |
| | (1) | (2) | (3) | (4) | (5) |
| | All immigrants | Skilled | Unskilled/Unknown qualification | EU | Non-EU |
| Xenophobic violence | -0.00006 (0.00007) | -0.00004 (0.00006) | -0.00025 (0.00015) | -0.00010 (0.00006) | 0.00001 (0.00002) |
| N | 2,778 | 2,778 | 2,778 | 2,778 | 2,778 |
| $R^2$ overall | 0.06 | 0.03 | 0.02 | 0.02 | 0.24 |
| $R^2$ within | 0.76 | 0.61 | 0.69 | 0.65 | 0.80 |
| $R^2$ between | 0.05 | 0.02 | 0.02 | 0.02 | 0.27 |
| | Fixed effects instrumental variable estimation | | | | |
| | All immigrants | Skilled | Unskilled/unknown qualification | EU | Non-EU |
| Xenophobic violence | -0.00144 (0.00107) | -0.00136** (0.00067) | -0.00102 (0.00203) | -0.00142 (0.00087) | -0.00004 (0.00030) |
| First stage | | | | | |
| IV supply vocational training | -0.61676** (0.26729) | | | | |
| IV foreign population | 0.05863*** (0.01683) | | | | |
| Kleibergen–Paap LM test (p value) | 18.65 (0.0001) | | | | |
| F Test of excluded instruments | 10.10 | | | | |
| Hansen J (p value) | 0.5976 | 0.3705 | 0.1245 | 0.6035 | 0.3057 |
| Anderson–Rubin Wald test (p value) | 4.58 (0.1013) | 6.53 (0.0381) | 4.21 (0.1220) | 5.94 (0.0514) | 1.07 (0.5854) |
| N | 2,772 | 2,772 | 2,772 | 2,772 | 2,772 |

All models include time-varying explanatory variables and region- as well as time-fixed effects to control for observed and unobserved factors. Robust standard errors in parentheses are clustered at the region level

* $p < 0.10$

** $p < 0.05$

*** $p < 0.01$.

Notes: IV supply vocational training: lagged (1 year) log ratio of vocational training positions to graduates demanding training. IV foreign population: lagged (9 years) log share of foreign population interacted with longitude.

statistically significant. This is in notable contrast to the results of the IV regression in the lower panel for skilled migrants. We find a statistically significant adverse effect for this group. The coefficient estimates for the average effect and the immigrants from other EU countries are even larger, but not precisely estimated. These IV estimates almost match the unconditional correlation. The coefficient in column (2) suggests that an increase in the rate of xenophobic violence by 1% relative to the remaining country reduces, on average, the immigration rate by 0.00144 percentage points or, in other words, leads to a decline in immigration of 0.014 foreign workers per 1,000 inhabitants.

The substantial differences between the fixed effects results and IV estimates might point to a measurement error that could severely affect the fixed effects estimation. The recording of politically motivated crime in Germany might give rise to a measurement error in the rate of xenophobic violence (see section 4.1). Fixed effects models are particularly prone to measurement errors, which may result in an attenuation bias. If a measurement error in the rate of xenophobic violence applies especially to the variation in time, the fixed effects estimation might not detect existing effects because identification makes use of this variation only [83, 84].

The discussion of the indicators for xenophobic attitudes in section 4.1 suggests that the share of right-wing votes and the measure of xenophobic violence might capture different aspects of hostile attitudes and may thus influence migration decisions differently. In our data, the correlation of the two variables is positive, but not particularly strong (correlation

coefficient: 0.46). To investigate whether xenophobic attacks and the support for right-wing parties have separate effects on the immigration rate, we also estimate models that include the two indicators simultaneously (see Table A7 in S7 Table). However, the sample size for this model strongly declines as the data availability of the two variables does not completely overlap. The results are qualitatively in line with our main results, i.e. the coefficients of the two variables are negative with the absolute size being clearly larger for the right-wing votes and increasing compared to the estimates in Table 1. In contrast, there are only minor changes for xenophobic violence. The effect of right-wing votes seems to be more robust than the impact of xenophobic violence, but the precision of estimates is affected by the reduction of the sample size and standard errors significantly increase as a result. Altogether, the results do not lend support to the assumption that right-wing votes and xenophobic violence capture different aspects of anti-immigrant attitudes which might affect migration behaviour separately.

The models in Tables 1 and 2 include the share of the foreign population to account for the influence of demographic characteristics of the region and amenities that might be linked to a large foreign population on migration decisions (see [5] for corresponding evidence on Germany). To consider the effects of local coethnic communities and differences in the distribution of ethnic groups across space, we extend our basic model by the share of the individuals' own nationality. Table 3 reports the results for both right-wing votes and xenophobic violence as well as the estimates of coethnic community effects. We focus on the IV results and the average effect for all migrant groups. The results in columns (1) and (2) indicate that the presence of conationals in fact increases the immigration rate of the respective group, which is in line with evidence by Nowotny and Pennerstorfer [85]. The impact of xenophobia is smaller than in Tables 1 and 2, but we still detect a statistically significant effect for the share of right-wing votes. The effect of xenophobic violence is, however, not precisely estimated. The estimates for the local size of coethnic communities indicate that a 1% increase in the population share of one's own nationality gives rise to an increase in the immigration rate of the corresponding group that ranges between 0.00015 and 0.00028 percentage points, i.e., 0.0015 to 0.0028 immigrants per 1,000 workers, depending on the specification.

The models in columns (3) and (4) also include an interaction between the share of one's own nationality and the corresponding xenophobia measure. The statistically significant coefficients of the interaction terms indicate that the adverse effect of xenophobic attitudes tends to increase as the size of the local ethnic community grows. This applies to both right-wing votes and xenophobic violence. We interpret this as possibly pointing to an amplifying impact of information on xenophobia, which might be transferred via local coethnic communities. Fig 3 shows the effects of right-wing votes and xenophobic violence for different local shares of coethnic groups, applying the range that we observe in our data. We display the elasticity of the immigration rate with respect to xenophobic attitudes. In the extended model, the change in the immigration rate with a 1% change in xenophobic attitudes is a function of the share of the own ethnic group ($\mu = \beta + \tau \cdot \ln group_{it-1}$). The significant interaction effects imply that the unfavourable impact of anti-immigration attitudes will rise over time as the size of the migrant population grows.

## 6. Discussion and conclusions

In view of the challenges of demographic change and the shortage of skilled workers, Germany, like many other industrialized countries, is dependent on the immigration of skilled workers from abroad. Regions in Germany that are particularly affected by these challenges are often also characterized by a high level of xenophobic attitudes, which may repel migrants. Regarding internal migration, previous studies provide confirming evidence. They show that

**Table 3. Ethnic communities—direct and interaction effects for the largest immigrant groups.**

| | Fixed effects instrumental variable estimation | | | |
|---|---|---|---|---|
| | **(1) Right-wing votes** | **(2) Xenophobic violence** | **(3) Right-wing votes** | **(4) Xenophobic violence** |
| Share of right-wing votes | -0.00041*** (0.00013) | | -0.00737** (0.00319) | |
| Rate of xenophobic violence | | -0.00010 (0.00006) | | -0.00150** (0.00067) |
| Interaction, own group | | | -0.00090** | -0.00017** |
| | | | (0.00039) | (0.00008) |
| Share of own group | 0.00028*** | 0.00015*** | 0.00017*** | 0.00018*** |
| | (0.00001) | (0.00001) | (0.00005) | (0.00002) |
| Share of foreign population | 0.00026*** (0.00005) | 0.00010*** (0.00003) | 0.00037*** (0.00009) | 0.00011*** (0.00003) |
| *IVs for xenophobia* | | | | |
| IV supply vocational training | -0.75106*** (0.06478) | -0.64236*** (0.08531) | -0.85847*** (0.07830) | -0.41082*** (0.10457) |
| IV low-skilled foreign workers | 0.01176*** (0.00117) | | | |
| IV foreign population | | 0.05940*** | | |
| | | (0.00561) | | |
| Instrument Interaction | | | 0.19702 | -0.22804 |
| | | | (0.12088) | (0.14053) |
| *IVs for interaction* | | | | |
| IV supply vocational training | | | 7.00879*** (0.59368) | 3.81593*** (0.81124) |
| Instrument Interaction | | | -2.55217*** | -0.29865 |
| | | | (0.85796) | (1.01323) |
| Kleibergen–Paap LM test | 211.66 (0.0000) | 176.22 (0.0000) | 11.11 (0.0009) | 21.02 (0.0000) |
| F Test of excluded instruments | 110.60 | 95.42 | 72.59 | 15.65 |
| | | | 80.32 | 14.19 |
| Hansen J (p value) | 0.7923 | 0.7347 | - | - |
| Anderson–Rubin Wald test (p value) | 10.87 (0.0044) | 3.18 (0.2036) | 10.31 (0.0058) | 6.22 (0.0446) |
| N | 10,471 | 30,354 | 10,471 | 30,354 |

All models include time-varying explanatory variables and region-group- as well as time-fixed effects to control for observed and unobserved factors. Robust standard errors in parentheses are clustered at the region-group level, * p < 0.10

** p < 0.05

*** p < 0.01.

Notes: IV supply vocational training: lagged (1 year) ratio of vocational training positions to graduates demanding training. IV low-skilled foreign workers: lagged (2 years) log share of low-skilled among foreign workers interacted with longitude. IV foreign population: lagged (9 years) log share of foreign population interacted with longitude.

immigrants avoid moving to places where they have to fear anti-immigrant attitudes [e.g., 15, 18].

We examine the influence of xenophobia on the choice of the first place of residence of employees from abroad in Germany. Thereby this study provides first evidence on the relationship between anti-immigrant attitudes and immigration for Germany. Secondly, we add new results to the literature which investigates the initial location choices of immigrants in destination countries and examine whether the adverse effects of xenophobic attitudes differ across skills groups and between EU and non-EU nationals. Differences in the migration behaviour due to anti-immigrant attitudes have rarely been considered in the literature so far. Moreover, in contrast to previous studies, we also examine whether the effect of xenophobic attitudes is influenced by the local size of coethnic communities and provide first evidence on an amplifying impact of large local communities. Our results are summarised in Table 4 in section 5. We find a negative effect of xenophobic attitudes on the immigration rate that is also

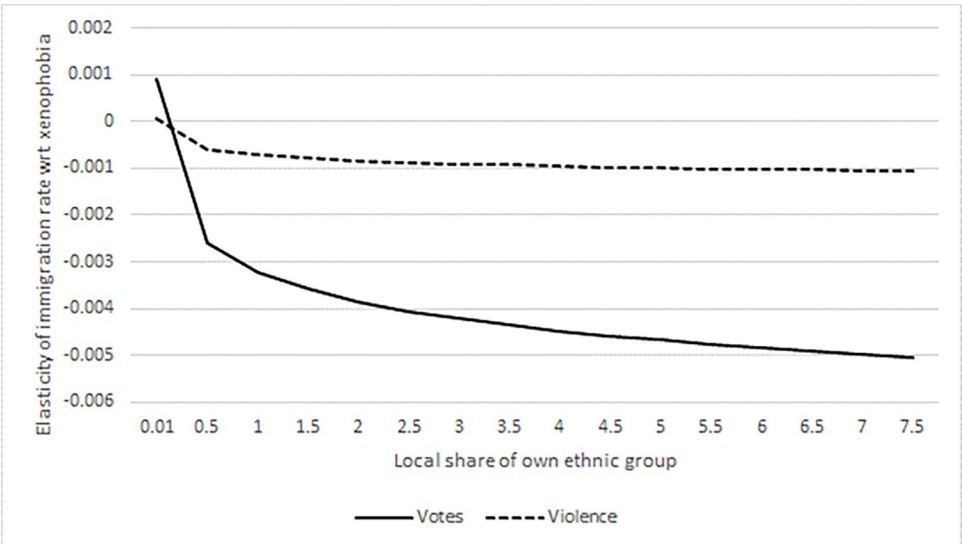

**Fig 3. Interaction effects: Elasticity of immigration rate with respect to xenophobia and local share of own ethnic group.** Source: own calculation.

statistically significant for right-wing votes, i.e., ceteris paribus, labour immigration is lower in regions characterized by relatively strong support for far-right parties and more frequent xeno-phobic violence. Hypothesis 1 can therefore be confirmed. However, the size of the adverse effects appears to be moderate, and evidence seems to be more robust for the support of right-wing parties than for xenophobic violence. The latter might be due to measurement errors caused by the reporting system of political motivated crime (see section 4.1).

Moreover, we find differences in the settlement behaviour of different groups of migrants. EU migrants seem to react more sensitively to regional right-wing vote shares than migrants from non-European countries. This might reflect a more flexible location choice due to the

**Table 4. Evidence on hypotheses.**

| Independent variable | | Dependent variable | Postulated effect | Hypothesis confirmed? |
|---|---|---|---|---|
| Xenophobic attitudes measured by. . . | . . .share of votes for right-wing parties | Immigration rate of a district region (H1) | - | Yes |
| | . . .xenophobic violence | | | ~ |
| | . . .share of votes for right-wing parties | Immigration rate of skilled compared to unskilled of a district region (H2) | - → - | No |
| | . . .xenophobic violence | | | Yes |
| | . . .share of votes for right-wing parties | Immigration rate of EU nationals compared to non-EU nationals (H3) | - → - | Yes |
| | . . .xenophobic violence | | | ~ |
| | . . .share of votes interacted with share of own group | Immigration rates (H4a) | – → - | No |
| | —xenophobic violence interacted with share of own group | | | No |
| | . . .share of votes interacted with share of own group | Immigration rates (H4b) | - → – | Yes |
| | —xenophobic violence interacted with share of own group | | | Yes |

Notes: ~ Sign or size of the coefficient in line with hypothesis but not precisely estimated.

complete freedom of movement for EU-workers. Moreover, the differences between EU and non-EU nationals may be traced back to the availability of information on anti-immigrant attitudes. Social phenomena such as right-wing radicalism are discussed in the media abroad. Since media coverage tends to decline with geographical distance [48, 49], EU citizens should be better informed about regional differences in xenophobic attitudes in Germany than people from outside the EU. However, Hypothesis 3 is only confirmed for right-wing votes while we find no statistically significant effect for xenophobic violence. A potential explanation is that recent electoral successes of right-wing parties have attracted a great deal of attention in international media.

In addition to traditional media, social media are gaining in importance as a source of information about possible settlement destinations [52, 53]. This source of information, together with personal contacts, could be responsible for the finding that the size of one's own ethnic group increases the negative influence of right-wing radicalism. Interestingly, a large group of one's own ethnicity does not seem to be primarily perceived as a protection against anti-immigrant behaviour, but rather acts as a broad source for collecting and transmitting information about local hostility. Thus, the results confirm hypothesis 4b while the dampening effect described by hypothesis 4a is rejected.

Regarding differences in the effects across migrant skill groups, our findings are ambiguous and our hypothesis 2 is only partly supported by the regression results. While the results for right-wing votes suggest that low-skilled foreign workers might be affected more strongly by xenophobic attitudes, there is a slight indication that xenophobic violence reduces the immigration of skilled workers more strongly. The greater sensitivity of low-skilled workers was not expected on the basis of the available studies [47, 86]. This raises the question whether skilled and unskilled immigrants perceive manifestations of anti-immigrant attitudes such as xenophobic violence and right-wing electoral success differently. We leave this as a subject of future research as our data provides no information on this topic.

The regional disparities in terms of demographic development in Germany are accompanied by parallel disparities in terms of economic prosperity, especially between East and West Germany. The immigration of skilled workers to lagging regions could help overcome the economic gap. Instead, regional differences in xenophobic attitudes and immigration seem to reinforce these disparities [87]. Moreover, an additional effect might be at work at the national level. In our study, we consider workers who decided to immigrate to Germany. However, a significant group of workers may have already decided against Germany as a destination country for fear of being confronted with right-wing radicalism. This is in line with recent evidence provided by Gorinas and Pytliková [16]. They show that a 10% increase in a country's measure of natives' propensity to discriminate against immigrants reduces the migrant inflow by 3.6% and that the group of interest in the present study, migrants who migrate for economic reasons and are active on the labour market, is even more sensitive to anti-immigrant attitudes.

If Avdiu [47] is right that skill transferability will increase over time due to globalism, migrants might become even more sensitive to anti-immigrant attitudes in the future and might more carefully select regions where they feel safe and welcome. Consequently, political effort is needed to promote a positive attitude and acceptance of immigration among the local population [12]. This is particularly true given that the AfD's strong electoral successes in the last elections were repeatedly seen as a sign that the right-wing ideologies and extreme positions are increasingly accepted by the middle of society [e.g., 88]–which in turn presumably increases the repellent effect on immigration.

There are still some open issues regarding the relationship between xenophobia and immigration from abroad that are left for future research. This applies to the differences across skill groups mentioned above, but also to the heterogeneity that we detect for EU and non-EU

nationals. While we discuss potential explanation, more detailed research is required to learn about the actual mechanism that give rise to the differences that we observe across groups of migrants. Another important open issue concerns the impact that the behaviour of employers may have. It is likely that in regions with high reservations against foreigners, employers reinforce xenophobic tendencies through discriminating recruiting behaviour. Recent findings by Henn and Hannemann [87] indicate that prevalent xenophobic attitudes seem to influence the recruitment behaviour of firms in the East German region of Thuringia. Thus, the negative effects of anti-immigrant attitudes on labour migration might be caused by both workers avoiding moving to unpleasant and frightening places and firms rejecting foreign workers or adapting exclusionary practices. More generally, it is important to learn more about the specific mechanisms underlying the adverse effects of xenophobia on migration.

## Supporting information

**S1 Table. A1: Variable definitions and data sources.**
(DOCX)

**S2 Table. A2: Summary statistics.**
(DOCX)

**S3 Table. A3: Interpretation of control variables.**
(DOCX)

**S4 Table. A4: Instrument variable results for xenophobic violence—controls and first stage.**
(DOCX)

**S5 Table. Instrument discussion, Tables 5A and 5B.**
(DOCX)

**S6 Table. A6: Unconditional correlations between immigration rate, share of right-wing votes and rate of xenophobic violence.**
(DOCX)

**S7 Table. A7: Fixed effects models with right-wing votes and xenophobic violence.**
(DOCX)

**S8 Table. A8: Summary statistics for the share of selected immigrant groups in total immigrant population.**
(DOCX)

## Acknowledgments

We thank the editor and two anonymous referees for helpful comments and suggestion. We are also grateful for comments received at the 2023 Congress of the European Regional Science Association (Alicante), at the 21. IMISCOE Annual Conference (Lisbon), and the 2023 winter seminar of the German-speaking section of the European Regional Science Association (Spital/Pyhrn). We also thank Juliane Kühn for providing excellent research assistance.

## Author Contributions

**Conceptualization:** Tanja Buch, Carola Burkert, Annekatrin Niebuhr.

**Data curation:** Tanja Buch, Carola Burkert, Stefan Hell, Annekatrin Niebuhr.

**Formal analysis:** Stefan Hell, Annekatrin Niebuhr.

**Methodology:** Annekatrin Niebuhr.

**Writing – original draft:** Tanja Buch, Carola Burkert, Annekatrin Niebuhr, Anette Haas.

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
