## [Decision Letter · Decision Letter 0]

15 Aug 2024

PONE-D-24-28637Does xenophobic behaviour influence migrant workers´ regional location choice?PLOS ONE

Dear Dr. Burkert,

Thank you for submitting your manuscript to PLOS ONE. After careful consideration, we feel that it has merit but does not fully meet PLOS ONE’s publication criteria as it currently stands. Therefore, we invite you to submit a revised version of the manuscript that addresses the points raised during the review process. **Both reviewers emphasize the need for a clearer definition and theoretical discussion of the main concepts (e.g., xenophobic behavior and attitude) early in the introduction, as well as a more explicit statement of hypotheses and their alignment with the literature. In addition, both reviewers suggest expanding the literature review, which I also concur that you need to extend by involving political science research, particularly on the subject of immigration attitudes and far-right vote choice in Germany. Finally, both reviewers highlight the importance of clearly displaying control variables in the regression models and refining the presentation of the results, including justifying methodological choices such as the use of logs and the inability to analyze right-wing votes and xenophobic violence simultaneously.**

We look forward to receiving your revised manuscript.

Kind regards,

Cengiz Erisen

Academic Editor

PLOS ONE

Journal Requirements:

Reviewers' comments:

Reviewer's Responses to Questions

**Comments to the Author**

1. Is the manuscript technically sound, and do the data support the conclusions?

Reviewer #1: Partly

Reviewer #2: Yes

2. Has the statistical analysis been performed appropriately and rigorously? 

Reviewer #1: Yes

Reviewer #2: Yes

3. Have the authors made all data underlying the findings in their manuscript fully available?

Reviewer #1: Yes

Reviewer #2: Yes

4. Is the manuscript presented in an intelligible fashion and written in standard English?

Reviewer #1: Yes

Reviewer #2: Yes

5. Review Comments to the Author

**Reviewer #1: **Strengths:

*The study addresses a timely and important topic, given the ongoing debates about immigration and xenophobia in many countries.

*The use of a unique panel dataset strengthens the credibility of the findings.

Suggestions for Improvement:

1. The paper could benefit from a clearer definition of "xenophobic behavior" early in the introduction. While it is operationalized through right-wing party votes and xenophobic violence, a theoretical discussion would provide better context.

2. The clear and directional hypothesis is not explicitly stated in the text. Explicitly stating the hypotheses as separate sentences and explaining why these hypotheses are used with references to the literature would strengthen the article. The lack of clearly stated hypotheses creates uncertainty about which hypothesis the concepts in the operationalization section are intended to test. The hypotheses should be clearly written, and it should be explicitly stated which concept measures the variable in each hypothesis.

3. In the literature section, the racial threat hypothesis and contact theory are compared, and it is mentioned that contact theory will be used. However, instead of building the literature review part on contact theory, it predominantly centers on xenophobia and attitudes towards immigrants literature. Since contact theory is used, the literature should be further strengthened with references to how contact, especially intimate contact, affects labor immigrants (both skilled and unskilled) and vice versa—how contact affects the xenophobic attitudes and attitudes towards immigrants of German citizens, as well as the attitudes of labor immigrants regarding their entry into the job market. Furthermore, expand the literature review to include more “contact” studies that have examined similar issues in other countries. This would help situate the findings within a broader international context and highlight the contribution of this study to the existing literature.

4. While the use of fixed effects and IV estimation is commendable, the paper should provide a more detailed justification for the choice of instruments and discuss their validity in depth. The section on instrumental variables and their explanation in the analysis seem a little bit complex for any reader. These parts could be simplified while retaining the main points.

5. The analysis of heterogeneous effects across skill groups is a valuable contribution. However, the findings are described as ambiguous. It would be helpful to delve deeper into these results to provide a more nuanced interpretation and possibly identify factors that contribute to the observed differences.

6. The control variables mentioned in Supporting Info 1 and Supporting Info 2 tables (also referenced in the text) are not shown in the regression models. There are sections in the analysis that refer to these control variables (e.g., labor market competition). Clearly displaying the effects of these control variables both in the tables and in the explanatory sections of the analysis will contribute to the readability of the article.

7. As stated in the first point, because the hypotheses are not clearly presented, the arguments in the discussion section about whether the hypotheses are supported or not seem somewhat vague.

8. Strengthen the conclusion by summarizing the key findings and their implications more explicitly. Highlight the study's contributions and suggest directions for future research.

Conclusion:

Overall, the article makes a significant contribution to understanding the impact of xenophobic behavior on migrant workers' regional location choices. By addressing the suggestions outlined above, the authors can further strengthen the clarity, generalizability, and contribution to the literature of their study.

**Reviewer #2:** Review of PONE-D-24-28637: Does xenophobic behaviour influence migrant workers´ regional location choice?

The authors show, on the basis of existing statistics, if regional share of right-wing votes in German counties and the extend of xenophobic violence affect immigrants’ choice of where to live in Germany. The paper is well written and the results are of high theoretical and practical relevance. I only have some suggestions for improving the readability of the paper:

Lines 16 ff lines 44ff: The authors use the terms “xenophobic attitudes” and “xenophobic behaviour” quite interchangeable. They should say, just in the beginning, if “xenophobic attitudes” relate to “right wing voting” and “xenophobic behaviour” to “xenophobic violence”. Personally, I would prefer the terms “right wing voting” and “xenophobic violence” because these describe what the authors use as indicators.

Line 53 or elsewhere: For the international readership the authors should say little bit more about what a German county is. See also line 283

Lines 74 - 84: I am astonished that the authors present a summary of their results already here. Unusual.

Lines 101: A word about the number of people living in West- and East-Germany would be informative.

122: Give a description of “political motivated crime” according to the “polizeiliche Kriminalstatistik”. Here or in the discussion section the authors should also discuss the low reliability and validity of the indicator (which might also help to understand some of the results related to this indicator). See also 308.

151: A summary of the up-to-date state of contact theory is Pettigrew, T. F., & Tropp, L. R. (2011). When groups meet: The dynamics of intergroup contact. Psychology Press.

316: Mentioning the percentages of votes for the different parties received in the last election would give an impression of the dominance of the AfD in this context.

324 ff: “In some studies, the variables are negatively correlated, which is attributed to an opportunity effect caused by the existence of right-wing parties [62].” A short explanation would be helpful.

330: The naming of the size of the correlation for the study at hand here is unusual. I recommend moving it to the results section.

336 f: Adding the size of the different groups would be helpful.

369 ff: Why did the authors use the LOGARITHMS of ratios?

478 ff: The authors should remind the reader why they could not analyse effects of right-wing votes and xenophobic violence simultaneously.

609: The authors refer to columns in tables. Printing the column number in the head of the tables would be helpful.

6. PLOS authors have the option to publish the peer review history of their article (what does this mean?). If published, this will include your full peer review and any attached files.

Reviewer #1: No

Reviewer #2: No

---

## [Author Response · Author response to Decision Letter 0]

4 Nov 2024

Dear referees,

we would like to thank you very much for the helpful and constructive comments on our manuscript -Does xenophobic behaviour influence migrant workers´ regional location choice?- We are grateful for the opportunity to revise and resubmit our paper and appreciate the detailed suggestions. We think that we were able to deal with the issues that were raised by you and have marked the fundamentally revised and added parts with tracked changes. In the attached document (241101_Response letter) we respond to your comments and describe in detail how we dealt with them. We first repeat each comment in italic, black script and then answer in blue script. 

Thank you.

---

## [Decision Letter · Decision Letter 1]

16 Dec 2024

Do xenophobic attitudes influence migrant workers’ regional location choice?

PONE-D-24-28637R1

Dear Dr. Burkert,

We’re pleased to inform you that your manuscript has been judged scientifically suitable for publication and will be formally accepted for publication once it meets all outstanding technical requirements. Congratulations! Within one week, you’ll receive an e-mail detailing the required amendments. When these have been addressed, you’ll receive a formal acceptance letter and your manuscript will be scheduled for publication.

Kind regards,

Cengiz Erisen

Academic Editor

PLOS ONE

Additional Editor Comments (optional):

Reviewers' comments:

Reviewer's Responses to Questions

**Comments to the Author**

1. If the authors have adequately addressed your comments raised in a previous round of review and you feel that this manuscript is now acceptable for publication, you may indicate that here to bypass the “Comments to the Author” section, enter your conflict of interest statement in the “Confidential to Editor” section, and submit your "Accept" recommendation.

Reviewer #1: All comments have been addressed

Reviewer #2: All comments have been addressed

2. Is the manuscript technically sound, and do the data support the conclusions?

Reviewer #1: Yes

Reviewer #2: Yes

3. Has the statistical analysis been performed appropriately and rigorously? 

Reviewer #1: Yes

Reviewer #2: Yes

4. Have the authors made all data underlying the findings in their manuscript fully available?

Reviewer #1: Yes

Reviewer #2: Yes

5. Is the manuscript presented in an intelligible fashion and written in standard English?

Reviewer #1: Yes

Reviewer #2: Yes

6. Review Comments to the Author

Reviewer #1: Review of the Revised Article: "Do xenophobic attitudes influence migrant workers' regional location choice?"

Evaluation of Revisions Based on Previous Suggestions

Clearer Definition of "Xenophobic Behavior"

In the first round, I suggested providing a theoretical discussion of xenophobic behavior in the introduction. The revised manuscript now includes a clearer definition, with an expanded theoretical explanation in the introduction. This improvement enhances the reader's understanding of how the concept is operationalized.

Explicit Hypotheses

Previously, I noted that hypotheses were not clearly stated. The revised manuscript now explicitly states directional hypotheses in a dedicated section, supported by references to the relevant literature. This change improves the clarity and structure of the study's aims.

Expansion of the Literature Review on Contact Theory

While the earlier version emphasized xenophobia and attitudes towards immigrants, I recommended integrating more studies on contact theory and its implications for labor immigrants. The authors have expanded the literature review to include references to contact theory, particularly regarding its effects on both skilled and unskilled labor immigrants. Additionally, comparative examples from other countries have been incorporated, situating the findings in a broader international context.

Justification for Instrumental Variables

The first round review highlighted the need for a detailed justification of the instrumental variables. In the revised manuscript, the authors have included a more comprehensive discussion of the validity and relevance of the instruments, making this section more accessible to readers while retaining its rigor.

Interpretation of Heterogeneous Effects

Previously, I suggested providing a deeper interpretation of the findings related to skill group differences. The revised manuscript offers a more nuanced discussion of these results, identifying potential contributing factors to the observed heterogeneity.

Control Variables in Regression Models

The lack of displayed control variables in regression tables was a significant issue. In the revised manuscript, the authors now present these variables in the main tables and discuss their effects in the analysis section. This change greatly improves the transparency and readability of the results.

Discussion Section and Hypotheses Clarity:

In the first round, I noted vagueness in the discussion section regarding the support for hypotheses. The revised version explicitly links the results back to the hypotheses, clearly stating whether they are supported or not, thus improving the coherence of the discussion.

Conclusion and Key Findings:

The earlier version lacked a strong conclusion summarizing the key findings and implications. The revised manuscript now provides a well-structured conclusion that highlights the study’s contributions and offers valuable suggestions for future research.

Additional Suggestion

Consider emphasizing the policy implications of the findings more prominently in the conclusion.

**The authors have made substantial improvements in response to the first-round suggestions. The revised manuscript is clearer, better structured, and more theoretically grounded. I now recommend this article for publication, provided the authors address the minor suggestion above to further refine the manuscript.

Reviewer #2: The authors responded to my forgoing recommendations adequately. xxxxxxxxxxxxxxxxxxxxxxxxxxxxxxxxxxxxxxxxxxxxxxxxxxxxxxxxxxxxxxx

7. PLOS authors have the option to publish the peer review history of their article (what does this mean?). If published, this will include your full peer review and any attached files.

Reviewer #1: No

Reviewer #2: No

---

## [Editor Report · Acceptance letter]

3 Jan 2025

PONE-D-24-28637R1 

PLOS ONE

Dear Dr. Burkert, 

I'm pleased to inform you that your manuscript has been deemed suitable for publication in PLOS ONE. Congratulations! Your manuscript is now being handed over to our production team.

Kind regards, 

on behalf of

Dr. Cengiz Erisen 

Academic Editor

PLOS ONE